# From bulk effective mass to 2D carrier mobility accurate prediction via adversarial transfer learning

Xinyu Chen[1], Shuaihua Lu[1], Qian Chen[1], Qionghua Zhou[1,2] ✉ & Jinlan Wang [1,2] ✉

Data scarcity is one of the critical bottlenecks to utilizing machine learning in material discovery. Transfer learning can use existing big data to assist property prediction on small data sets, but the premise is that there must be a strong correlation between large and small data sets. To extend its applicability in scenarios with different properties and materials, here we develop a hybrid framework combining adversarial transfer learning and expert knowledge, which enables the direct prediction of carrier mobility of two-dimensional (2D) materials using the knowledge learned from bulk effective mass. Specifically, adversarial training ensures that only common knowledge between bulk and 2D materials is extracted while expert knowledge is incorporated to further improve the prediction accuracy and generalizability. Successfully, 2D carrier mobilities are predicted with the accuracy over 90% from only crystal structure, and 21 2D semiconductors with carrier mobilities far exceeding silicon and suitable bandgap are successfully screened out. This work enables transfer learning in simultaneous cross-property and cross-material scenarios, providing an effective tool to predict intricate material properties with limited data.

Data-driven machine learning (ML) has succeeded in rapidly predicting material properties for data-rich systems such as perovskites[1,2], alloys[3,4], and catalysis[5,6]. Properties including formation energy[7], stability[8], and bandgap[9] can be predicted almost instantaneously, significantly accelerating material discovery compared with the traditional trial-and-error approach using experiments and simulations[10]. ML heavily relies on the quantity and quality of training data as a data-driven approach. However, high-fidelity data for complex properties are often insufficient, compromising its prediction accuracy[11,12]. In addition, data insufficiency may also cause incompleteness, which can lead to the ML model constantly suffering from overfitting and poor generalizability[13].

Transfer learning is a machine-learning technique that can improve the performance of learners on small datasets (target domain) by transferring knowledge from different but large datasets (source domain). It has been considered a very promising approach to address the data scarcity challenge in ML-assisted material design[14,15]. For example, Liu et al. successfully predicted phonon properties of bulk semiconductors by training on 1245 electronic bandgaps and fine-tuning on 124 phonon bandgaps[16]. Similarity, Li et al. accurately predict the formation energy of perovskite oxides by training on 5329 spinel oxides and finetuning on 855 perovskite oxides[17]. However, current transfer learning applications are either between different properties with the same materials (cross-property) or between different materials with the same property (cross-material)[18–25]. This is owing to that the effectiveness of transfer learning is closely related to the difference between the source and target domain, and if the domain difference is too large, it will not be effective and may give poorer predictions, i.e., negative transfer[26].

In practical applications, the problem of data scarcity becomes even more pronounced, as our extensive databases typically only cover fundamental properties of widely-used materials. Yet, our focus is often

[1]Key Laboratory of Quantum Materials and Devices of Ministry of Education, School of Physics, Southeast University, Nanjing, China. [2]Suzhou Laboratory, Suzhou, China. ✉e-mail: qh.zhou@seu.edu.cn; jlwang@seu.edu.cn

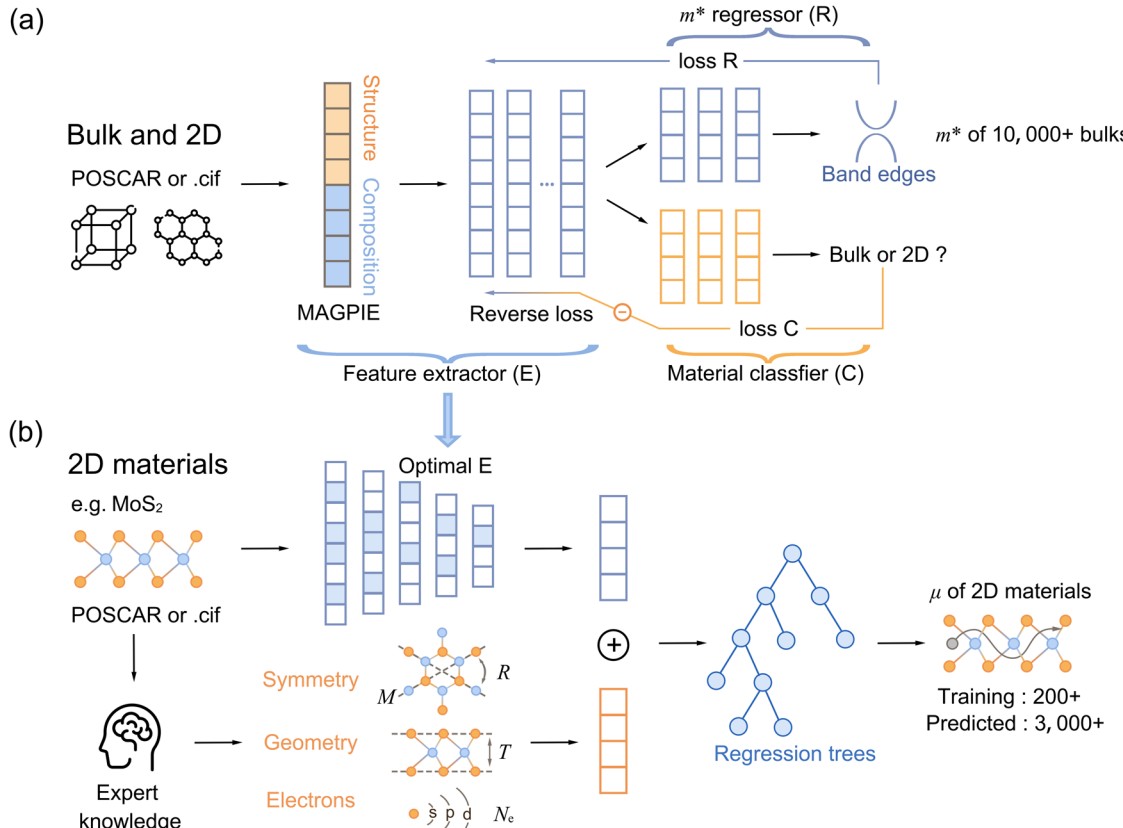

**Fig. 1 | Schematic of adversarial transfer learning from bulk effective mass to 2D carrier mobility. a** Adversarial transfer learning. Both bulk and 2D materials are first transformed into feature vectors based on their structures and compositions using materials materials-agnostic platform for informatics and exploration (MAGPIE), and then a multi-layer perceptron is used to extract features. The extracted features are used for two tasks: bulk effective mass regression and material classification. When backpropagating the regressor (R) loss and the reversed classifier (C) loss, the feature extractors are trained to extract only shared features between the target and source domains. E and C are trained iteratively until C can no longer identify bulk and 2D materials. **b** 2D carrier mobility prediction with expert knowledge. Using H-MoS$_2$ as an example, features related to effective mass are extracted by E from its POSCAR, moreover, based on expert knowledge, features like mirror ($M$) and rotational ($R$) symmetry, layer thickness ($T$), and valence electron number ($N_e$) are also extracted. With this hybrid feature vector, final predictions on 2D carrier mobility are given by well-trained regression tree models.

on a particular category of materials, for which we strive to predict their more complex properties. Carrier mobility in atomically thin 2D semiconductors is such a typical example. 2D materials with suitable bandgap and high carrier mobility are expected to facilitate the continued transistor scaling[27,28]. However, the evaluation of carrier mobility is a costly process that often requires extensive density functional theory calculations, as a result, the available data is very limited[29,30]. In addition, 2D materials themselves are recent additions to the material family which also lacks sufficient data. In contrast, bulk materials have been studied for a much longer period and have rich data available, including diverse properties, in which the effective mass is believed to be closely related to carrier mobility. Naturally, we hope to utilize bulk effective mass data to enhance the prediction of 2D carrier mobility. However, owing to the diversity of 2D material structures and the complexity of their properties, simultaneous cross-material and cross-property transfer learning poses a greater challenge.

To achieve such simultaneous cross-material and cross-property transfer learning, we propose a hybrid framework that combines domain adversarial training and expert knowledge. The domain adversarial training method was first introduced in the realm of computer vison to learn common knowledge between different images[31]. Here, we employ a similar adversarial training concept to acquire common knowledge between different materials, meanwhile, we incorporate a priori knowledge of chemistry to better describe the uniqueness of material property. Successfully, 2D carrier mobility can be predicted within an order of magnitude by simply inputting crystal

structure files, and 21 semiconductors with ultrahigh carrier mobility ($> 10^4\,cm^2/V\cdot s$) and suitable bandgap are screened out. This successful knowledge transfer across different materials and properties shows the potential to fully utilize existing data, which may be an effective tool for material design with limited data.

## Results
### Hybrid transfer learning framework
Our transfer learning framework consists of two main components. The first part utilizes adversarial transfer learning (ATL) to extract shared features from both bulk materials and 2D materials, as shown in Fig. 1(a). The adversarial transfer learning is composed by three multi-layer perceptron (MLP) models: a feature extractor, an effective mass regressor, and a data source classifier. The feature extractor transforms initial input features into a low-dimensional vector using materials agnostic platform for informatics and exploration (MAGPIE)[32]. This extractor can be applied to both bulk and 2D materials; initially, without any constraint, the output of the extractor is a random number. Meanwhile, the extracted features are also used to train the bulk material effective mass regressor, and the regression loss is backpropagated to optimize the feature extractor. At this stage, the feature extractor learns the knowledge of effective mass and provides features closely related to it. In contrast to the standard approach, we not only train an effective mass regressor but also an additional data source classifier. This classifier is designed to determine whether the features are extracted from bulk or 2D materials and tell the feature extractor

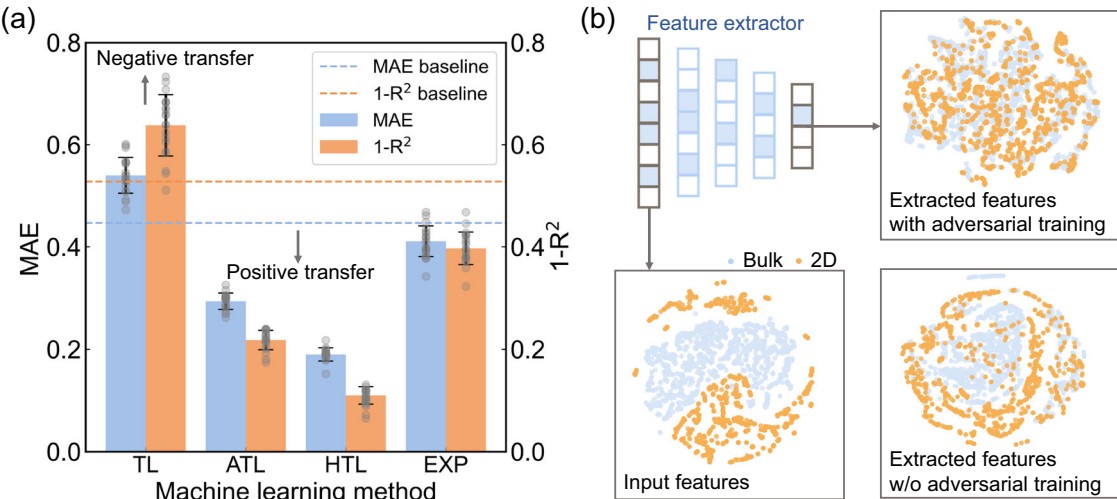

**Fig. 2 | Effective knowledge transfer enabled by adversarial transfer learning. a** Performance comparison of different machine learning methods includes transfer learning without adversarial training (TL), adversarial transfer learning (ATL), hybrid transfer learning with adversarial training and expert knowledge (HTL) and machine learning with only expert knowledge (EXP). The model performance is evaluated by mean absolute error (MAE) and coefficient of determination ($R^2$) through 20-fold cross-validation, and the data are presented as mean values +/− standard deviation of 20 samples. The baseline (dash line) is set to direct prediction from initial features without transfer learning. **b** Illustration of the latent feature space of input features and extracted features with different transfer learning methods using t-distributed stochastic neighbor embedding (t-SNE). The orange and blue dots represent 2D and bulk materials respectively, while the top left panel serves as an illustration of the trained feature extractor. Source data are provided as Source Data files.

the common features between both types of materials. We achieve this by backpropagating the reversed classification loss to the feature extractor, and iteratively training both extractor and classifier until the classifier can no longer identify the data source. During the training iterations, the feature extractor is trained to fool the classifier about the data source, while the classifier is trained to discriminate between them. Hence, this process is referred to as adversarial training.

The second component of our model involves the embedding of expert knowledge and provides a direct prediction of 2D carrier mobility, as illustrated in Fig. 1(b). The necessity of incorporating expert knowledge lies in the fact that the adversarial approach only ensures the extraction of common knowledge, but it lacks the description of the uniqueness of target materials and their properties. This is particularly critical in cases like 2D materials, where many interesting properties stem from their unique structures. Therefore, in addition to the features extracted by transfer learning, we add features from lattice symmetry, crystal geometry, and electronic properties to describe the unique 2D structures and their electronic behavior. From the perspective of deformation potential theory, we speculate that these features are closely related to carrier mobility. For example, the symmetry can affect phonon vibration mode, while the thickness is related to elastic modulus and reflects the strength of electron-phonon coupling, thus contributing to carrier mobility. The electronic features, such as electronegativity and valence electron distribution, are also regarded as important for carrier transport. The full feature list and description are presented in Supplementary Table 2. It is worth noting that these added features can be directly taken from the structure files without additional density functional theory (DFT) calculation, which is critical for quick-and-direct prediction.

To demonstrate the impact of adversarial training and expert knowledge on the performance of cross-material transfer learning, we conducted comparative tests, as shown in Fig. 2(a). When transfer learning is applied without adversarial training, the extracted features perform worse than MAGPIE (baseline), which is a typical negative transfer. This indicates that although the extracted features work in the source domain, they may not necessarily be helpful in the target domain, especially when the source and target domains are different materials with different properties. However, with the help of

adversarial training, common knowledge between bulk and 2D materials is captured, and negative transfer is alleviated. It is also essential to recognize that many appealing properties of 2D materials stem from their unique structure. Therefore, additional features based on expert knowledge describe their special structures, which complements the knowledge acquired from the bulk materials and leads to more accurate predictions. This demonstrates the effectiveness and importance of leveraging adversarial training and expert knowledge to enhance the transfer learning performance.

To further investigate how adversarial transfer learning works, we utilized t-distributed stochastic neighbor embedding (t-SNE) to visualize the output differences between our transfer learning model with and without adversarial training, as shown in Fig. 2(b). The input feature space of 2D and bulk materials is separated, which implies that the overlap is small, making it challenging to transfer knowledge from bulk to 2D materials. We can see that most of the bulk and 2D materials are still separated after the transfer learning without adversarial training. However, after incorporating adversarial training, the extracted features are no longer able to distinguish between bulk and 2D materials, it can be observed that the two types of materials are mixed together in the t-SNE plot. This indicates that the features extracted by adversarial transfer learning are shared by both bulk and 2D materials, therefore, improving the effectiveness of cross-material transfer learning.

## Carrier mobility prediction and model interpretation

Our hybrid transfer learning framework has been applied to three tasks related to 2D carrier mobility. Figure 3(a) displays the prediction accuracy of carrier mobility under deformation potential theory (DPT), with $R^2$ values of 0.88 and 0.90 for the average electron and hole mobility, respectively, and a MAE of 0.19 for both mobilities. Importantly, our trained model only requires the crystal structure file as input when making carrier mobility predictions. This streamlined approach ensures the usability and efficiency. Compared to DFT-based mobility calculation, our approach is five orders of magnitude faster as shown in Supplementary Fig. 10. These results demonstrate that our models can provide accurate and efficient estimation of overall carrier mobility. In addition, feature importance analysis in Fig. 3(d) reveals

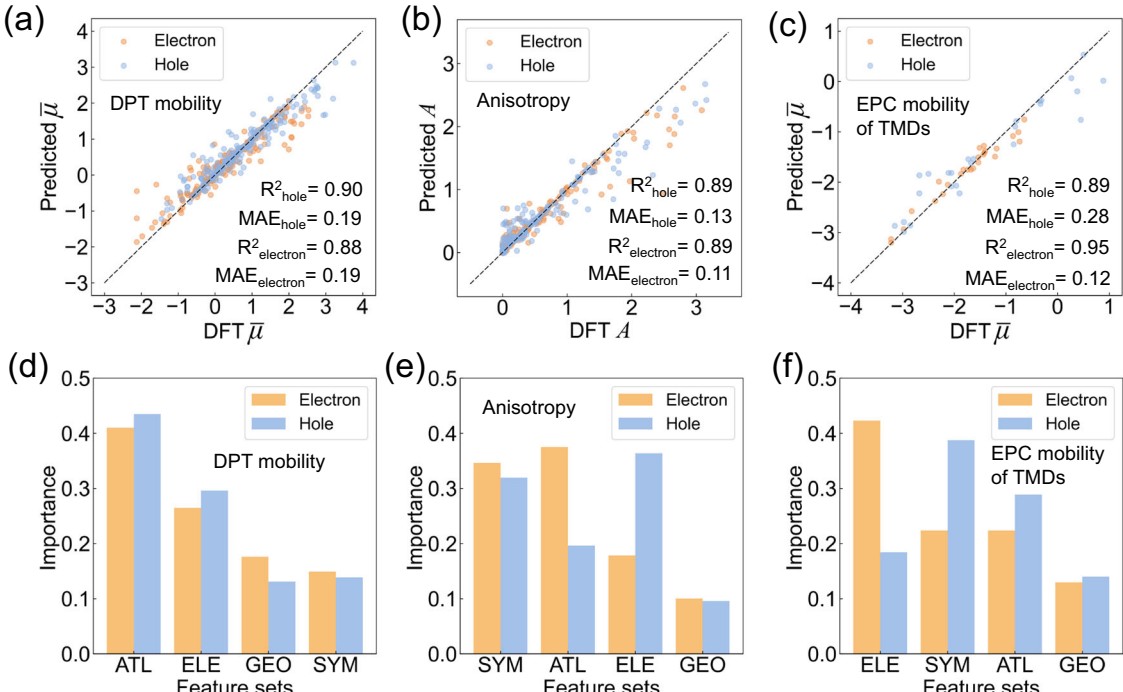

**Fig. 3 | Model performance and interpretation.** Prediction accuracy on (**a**) average carrier mobility (defined in Formula 2) under deformation potential theory (**DPT**); (**b**) mobility anisotropy (defined in Formula 3); (**c**) average carrier mobility of transition metal dichalcogenides (TMDs) via solving electron-phonon coupling (EPC) matrix. The model performance is assessed using mean absolute error (MAE) and coefficient of determination ($R^2$), with the corresponding values displayed as text. The feature importance of various feature sets, including adversarial transfer learning learned features (ATL), electronic features (ELE), symmetry features (SYM), and geometry features (GEO), is analyzed for models predicting (**d**) average DPT carrier mobility, (**e**) mobility anisotropy, and (**f**) average EPC carrier mobility of TMDs. The orange and blue colors represent electrons and holes, respectively, and the corresponding features in each feature set can be found in Supplementary Table 2. Source data are provided as Source Data files.

that the most important predictors for average carrier mobility are ATL features, closely related to effective mass. Furthermore, electronic features such as valence electron distribution, which characterize the electronic distribution of materials, are also crucial predictors. Our models also accurately predict mobility anisotropy, as evidenced by the $R^2$ scores of 0.89 and MAE of 0.11 and 0.13 for electron and hole, respectively, as shown in Fig. 3(b). Symmetry features, such as space group and mirror symmetry, play a more prominent role in determining the mobility anisotropy, as shown in Fig. 3(e), compared to that of ATL features. These results indicate that mobility anisotropy is mainly determined by the symmetry of the material.

Recently, a more accurate method to estimate carrier mobility was developed by solving the electron-phonon coupling (EPC) matrix, which provides a valuable re-evaluation of the carrier mobility of common 2D materials and gives more consistent results with experimental measurements[33]. We further tested our method of predicting EPC mobilities of transition metal dichalcogenides (TMDs), which show high prediction accuracy as illustrated in Fig. 3(c). The $R^2$ score reaches 0.95 for electrons and 0.89 for holes, which is similar to DPT mobility prediction. Differently, the feature importance in Fig. 3(f) suggests that ATL features depicting effective mass are less important than electron features and symmetries in this case. This is in line with the finding that effective mass shows no obvious correlation with carrier mobility for 2D TMDs[34]. Despite the limited structure types hindering its extrapolation, our method still gives very good accurate predictions. With increasing amounts of high-accuracy data, it can be useful in the future. Nevertheless, this consistency and maintained accuracy prove that our method can provide robust predictions with great generalization ability.

With the well-trained model, we can screen out 2D candidates for high carrier mobility and proper bandgap that is comparable to silicon.

As shown in Fig. 4(a), 9115 2D materials were collected from two open-source databases, and a de-duplication process was carried out based on their formula and space group. Then we removed all metals, leaving 4266 semiconductors, of which 3109 are thermodynamically stable. Considering that silicon has a bandgap of around 1 eV, we selected 869 semiconductors with similar bandgaps ranging from 0.5 eV to 1.5 eV. Finally, the trained model was applied to predict their average carrier mobility $\bar{\mu}$, and 21 materials with electron or hole mobility higher than $10^4$ cm²/V·s were screened out. We further validate the accuracy of our ML model by DFT calculations based on effective mass approximation and deformation potential approximation, which provide reliable estimations of carrier mobility at an acceptable computational cost. As shown in Supplementary Fig. 9, for both electron and hole mobility, our ML model gives consistent predictions with the $R^2$ score above 0.82, and the MAE values below 0.22, demonstrating the great predictive ability of our ML model. Note that among the screened 21 materials, some have already been synthesized or even been experimentally validated to have high mobility, such as $In_4Se_3$ and $Nb_2SiTe_4$[35–37].

Figure 4(b) shows the element and crystal system distribution of selected materials with high carrier mobility. The most frequent elements belong to p-block, and the most common crystal syngony are parallelogram, orthorhombic, and rhombic syngony. To gain a deeper understanding about why these materials possess high carrier mobility, we conducted partial dependence analysis based on Shapley additive explanation (SHAP) values, as shown in Fig. 4(c–f). Regarding the elemental features, an increase in the p-valence electrons fraction is positively correlated with carrier mobility (Fig. 4(c)), consistent with the element distribution results in Fig. 4(b). Moreover, Fig. 4(d) shows that the smaller the difference in electronegativity, the more positive the contribution to carrier mobility. This may be because smaller

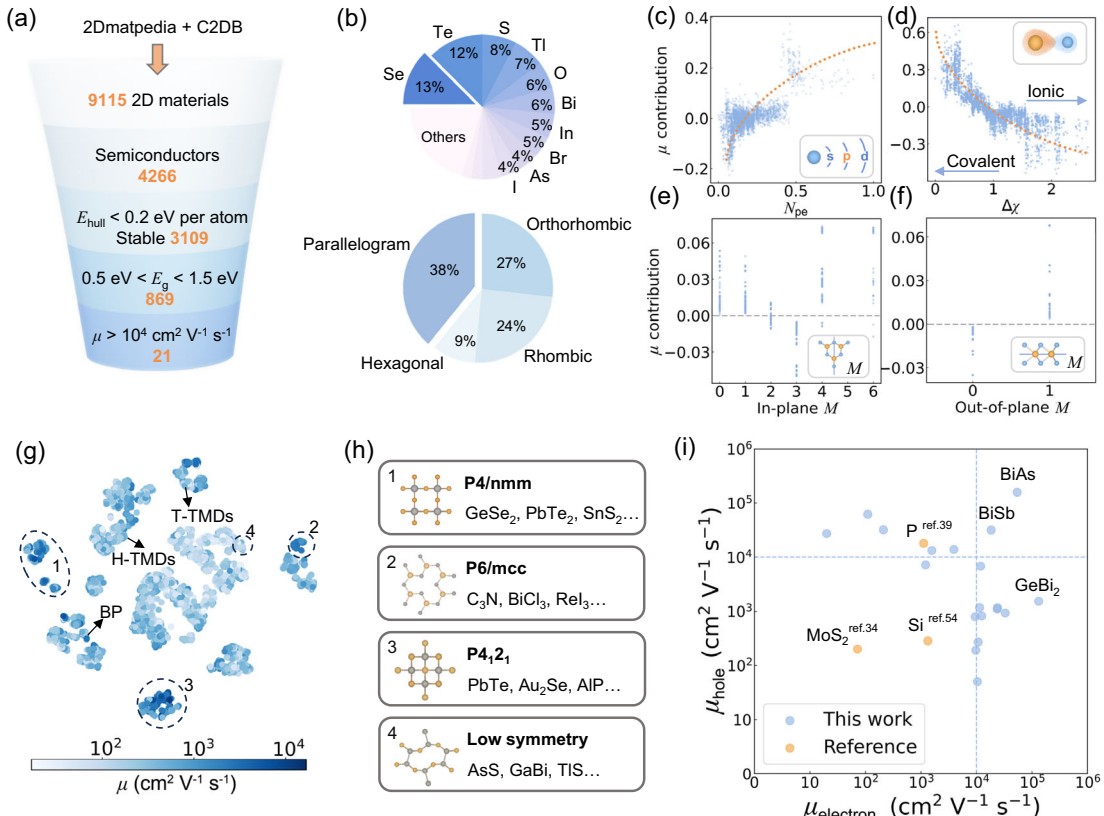

**Fig. 4 | Rational discovery of 2D semiconductors with high carrier mobility. a** A schematic illustration of the stepwise screening framework for the determination of promising 2D semiconductors with high carrier mobilities. **b** Partial dependence plots illustrate the mobility contribution of key factors, including (**c**) $p$ valence electron fraction ($N_{pe}$), (**d**) electronegativity difference ($\Delta\chi$), (**e**) in-plane (**f**) out-of-plane mirror symmetry ($M$) as evaluated by Shapley additive explanation (SHAP) with corresponding feature illustrations in the inserts. For a comprehensive view, the full SHAP plot is provided in Supplementary Fig. 6. **g** The t-distributed stochastic neighbor embedding (t-SNE) visualization of latent feature space. The scatter plot uses color to indicate the electron mobility of 2D semiconductors, with materials having high mobility selected within dashed line circles. Some popular 2D materials are also pointed out such as T-phase and H-phase transition metal dichalcogenides (T-TMDs and H-TMDs), and black phosphorus (BP). The hole mobility can be found in Supplementary Fig. 7. **h** The representative space group, crystal structure prototype, and examples of high mobility 2D semiconductors. **i** Electron and hole mobility of the ML screened 2D materials. Some common semiconducting materials are also plotted as orange dots for comparison. Source data are provided as Source Data files.

electronegativity difference tends to facilitate the formation of covalent bonds, in which the electrons are more free and easier to move, resulting in higher mobility. Structural features such as mirror symmetries are also found to correlate to carrier mobility. As shown in Fig. 4(e–f), materials with three in-plane mirror operations, such as TMDs, generally exhibit lower electron mobility, while materials with out-of-plane mirror operations, such as hexagonal boron nitride, have higher electron mobility. This may be because the mirror symmetry restricts some specific phonon vibration modes, which weakens the electron-phonon coupling and results in higher carrier mobility[38]. It is worth noting that although the trend regarding symmetry is obvious, the overall impact of symmetry on the model is not as significant as elemental features according to SHAP values. Therefore, in materials with different compositions, the effect of symmetry can be easily masked. Moreover, research on the correlation between symmetry and mobility is still limited, and further study based on phonon vibration modes is needed to provide deeper insights.

Figure 4(g, h) presents the distribution of carrier mobility in 2D semiconductors using t-SNE. Space groups including P6/mcc, P4/nmm, P4₁2₂, and some low symmetry systems are observed to have high carrier mobility. A significant trend is that the majority of 2D semiconductors exhibit higher electron mobility than hole mobility, except for some low symmetry structures where the hole mobility is higher, the t-SNE plot of hole mobility is shown in Supplementary Fig. 7. Further DFT calculations show that the effective mass of

electrons and holes in these systems are similar, while the deformation potential exhibits significant differences, as listed in Table 1 and Supplementary Table 3. Therefore, the high hole mobility in these low symmetry structures may be influenced by different phonon scattering mechanisms for electrons and holes. Although the materials selected in this study exhibit higher charge carrier mobility than traditional 2D semiconductors such as MoS₂ and black phosphorus, a considerable fraction of these materials have bandgaps that are either too large or too small for application in the semiconductor industry. However, these materials may still have potential applications in fields such as catalysis and photovoltaic applications. Figure 4(i) presents the electron and hole carrier mobility of the screened materials with proper bandgap around 1 eV, most materials have high mobility of only one carrier type, which are promising candidates for p-type or n-type semiconductors. Notably, BiAs and BiSb possess both high electron and hole mobility, serving as the compelling choice for complementary logic devices.

## 2D semiconductors with high carrier mobility

To gain insight into the mechanisms underlying the high mobility of these 2D semiconductors, we performed an electronic structure analysis of two representative structures (group V AB and group IV-V AB₂), as shown in Fig. 5. We also summarized their carrier mobility, bandgap, effective mass and deformation potential in Table 1 and Supplementary Table 3. Group V AB, they have a structure similar to black

**Table 1 | Calculated carrier mobility, effective mass and deformation potential for the top 10 materials with the highest carrier mobility**

| Material | Space group | $E_g$ (eV) | carrier | $E_x$ (eV) | $E_y$ (eV) | $m^*_x$ ($m_0$) | $m^*_y$ ($m_0$) | $\mu_x$ ($10^3$ cm$^2$ V$^{-1}$ s$^{-1}$) | $\mu_y$ |
|---|---|---|---|---|---|---|---|---|---|
| BiSb | P2$_1$ | 0.52 | e | 6.26 | 2.82 | 0.03 | 0.07 | 18.35 | 8.05 |
|  |  |  | h | 8.23 | 1.91 | 0.04 | 0.07 | 17.81 | 31.70 |
| BiAs | P2$_1$ | 0.55 | e | 6.81 | 1.82 | 0.05 | 0.08 | 10.09 | 53.96 |
|  |  |  | h | 8.24 | 1.36 | 0.04 | 0.06 | 10.35 | 158.11 |
| GeBi$_2$ | Pmc2$_1$ | 0.57 | e | 5.53 | 0.24 | 0.07 | 0.31 | 3.36 | 131.88 |
|  |  |  | h | 1.93 | 2.38 | 0.33 | 0.91 | 1.53 | 0.13 |
| Sb$_2$OSe$_2$ | P1 | 0.62 | e | 6.96 | 5.89 | 0.14 | 0.03 | 2.74 | 10.70 |
|  |  |  | h | 3.45 | 3.74 | 0.73 | 0.41 | 0.27 | 0.27 |
| Pb$_2$Se$_6$ | P1 | 0.74 | e | 0.55 | 1.95 | 0.19 | 0.73 | 23.48 | 0.22 |
|  |  |  | h | 2.11 | 0.58 | 0.36 | 1.02 | 0.25 | 1.10 |
| GeSb$_2$ | Pmc2$_1$ | 0.78 | e | 6.76 | 0.64 | 0.14 | 0.33 | 0.88 | 11.35 |
|  |  |  | h | 1.79 | 4.21 | 0.40 | 1.52 | 1.17 | 0.02 |
| AgI | Pmma | 0.85 | e | 1.64 | 1.08 | 0.29 | 0.30 | 4.64 | 9.47 |
|  |  |  | h | 1.99 | 1.85 | 0.74 | 0.28 | 0.79 | 0.31 |
| BiISe | Pmmn | 1.10 | e | 9.04 | 1.63 | 0.09 | 0.52 | 0.86 | 3.91 |
|  |  |  | h | 1.68 | 5.22 | 0.16 | 0.29 | 13.90 | 0.69 |
| In$_4$Se$_3$ | Pmn2$_1$ | 1.13 | e | 4.11 | 1.05 | 0.31 | 0.13 | 0.81 | 23.84 |
|  |  |  | h | 0.72 | 4.16 | 2.48 | 0.13 | 1.16 | 0.54 |
| AsS | P2$_1$/c | 1.20 | e | 11.52 | 2.74 | 2.89 | 0.39 | 0.01 | 0.11 |
|  |  |  | h | 0.32 | 5.32 | 0.45 | 0.13 | 61.91 | 0.37 |
| PbTe | P4/nmm | 1.26 | e | 1.14 | 1.14 | 0.14 | 0.14 | 32.78 | 32.78 |
|  |  |  | h | 6.44 | 6.44 | 0.15 | 0.15 | 0.93 | 0.93 |
| Ga$_2$Te$_3$ | Pc | 1.48 | e | 2.81 | 2.63 | 0.09 | 0.06 | 6.81 | 10.41 |
|  |  |  | h | 4.89 | 5.66 | 0.65 | 0.45 | 0.05 | 0.05 |

All materials are converted to orthogonal lattice, in which directions are labeled as *x* and *y*. For comparison, the electron and hole mobility in silicon are 1331 cm$^2$V$^{-1}$s$^{-1}$ and 283.5 cm$^2$V$^{-1}$s$^{-1}$ respectively[54]. Their crystal and electronic structures are given in Supplementary Fig. 11–16, and the structure files are provided in Supplementary Data 1.

phosphorus, as shown in Fig. 5(a)[39]. However, with different elements substituted, their band edges shift from the Γ point to a point within the Y-Γ high-symmetry path. Moreover, the shape of the shifted band edges is sharper, indicating smaller effective masses and higher carrier mobilities as shown in Fig. 5(b, c). These findings suggest that bandgap engineering through elemental substitution can be an effective approach to enhance carrier transport, and the elements within the p-block may be good choices according to the above partial dependence analysis. For the group IV-V AB$_2$ semiconductors, as shown in Fig. 5(d), they exhibit strong structural anisotropy, which leads to remarkable electronic anisotropy, as shown in Fig. 5(e, f). Local magnifications of conduction band minimum and valence band maximum reveal obvious differences in effective mass along orthogonal directions. Specifically, the effective mass is smaller along a lattice direction, interestingly, the deformation potential is far smaller along b direction than a direction (see Table 1). Consequently, the mobility is higher along the b direction, which indicates that the electron-phonon interaction plays a decisive role in this system. In addition, the a, b plane also show a symmetry difference. Specifically, the a direction exhibits mirror symmetry, while the b direction does not, further implying a potential correlation between symmetry and mobility. This highlights the importance of further investigation on how symmetry affects carrier mobility, as well as the potential for modulating carrier mobility through symmetry-protection or symmetry-broken structural engineering.

## Discussion

In summary, we have developed a hybrid transfer learning method that combines adversarial training and expert knowledge to enable effective knowledge transfer across different materials and different properties. As a compelling demonstration, this method has been applied 2D materials and achieved rapid and accurate predictions of carrier mobility by utilizing the big data of bulk effective mass. Notably, such mobility prediction only necessitates crystal structures as input, yet maintains accuracy comparable to DFT calculations but at a speed five orders of magnitude faster. Moreover, 21 2D semiconductors with ultra-high carrier mobility far exceed silicon have been screened out from 4266 candidates. The success of this method lies in the incorporation of adversarial training and expert knowledge, which effectively captures similarity among diverse materials while also characterizing the distinctive attributes of target materials and properties. Therefore, it facilitates simultaneous cross-material and cross-property transfer learning, enhancing the predictive capabilities and reliability of the model. This study provides a widely applicable strategy for addressing data scarcity in ML-assisted material design.

Nevertheless, the effectiveness of this approach for systems with higher degrees of dissimilarity, such as from ordered crystal to disordered materials like alloy, remains untested and may not be as successful. It may require an improved adversarial training approaches or borrowing generative adversarial network methods from inverse design[40–43]. Another significant challenge in this field is how to select the most appropriate source-domain tasks from various options available. This is especially important since the amount and diversity of materials data is constantly expanding. This highlights the need for further research to explore how to uncover the correlations between different materials and properties, which guides source task selection while also improves the interpretability in transfer learning.

## Methods

**Machine learning:** The transfer learning framework is composed of three parts: a feature extractor, a property regressor, and a data source discriminator. All of these parts utilize multi-layer perceptron (MLP) models, which are built and trained under the PyTorch[44] framework. To optimize the MLP hyperparameters, including the number of layers and neurons per layer, we employed a random search method, as depicted in Supplementary Fig. 3, 4. Then, the extracted features were fed into a gradient boosting tree (XGBoost) model to predict carrier mobility and its anisotropy. Other models, such as kernel ridge regression (KRR) and least absolute shrinkage and selection operator (LASSO) were also tested, and the tree model under the XGBoost framework gives the best performance, as illustrated in Supplementary Fig. 5, the optimized hyperparameters are given in Supplementary Table 1.

**Model interpretation:** To interpret the machine learning models, we used Shapley additive explanation (SHAP[45]), based on game theory by Lloyd Shapley. This method produces SHAP values that reflect the positive or negative impact of each feature in each sample on the prediction results, providing deeper interpretation capabilities for complex ML models. In this study, we assessed the importance of feature using the mean absolute SHAP values, which describe the average impact of each feature. Due to the relatively small dataset, which may lead to increased randomness, we used 20-fold cross-validation to assess the model performance.

**High-throughput calculations:** All high-throughput calculations were performed using a self-developed Python script within the framework of density functional theory (DFT) implemented in the Vienna ab initio Simulation Package (VASP[46]). Specifically, the electron-electron interactions were handled using a general gradient

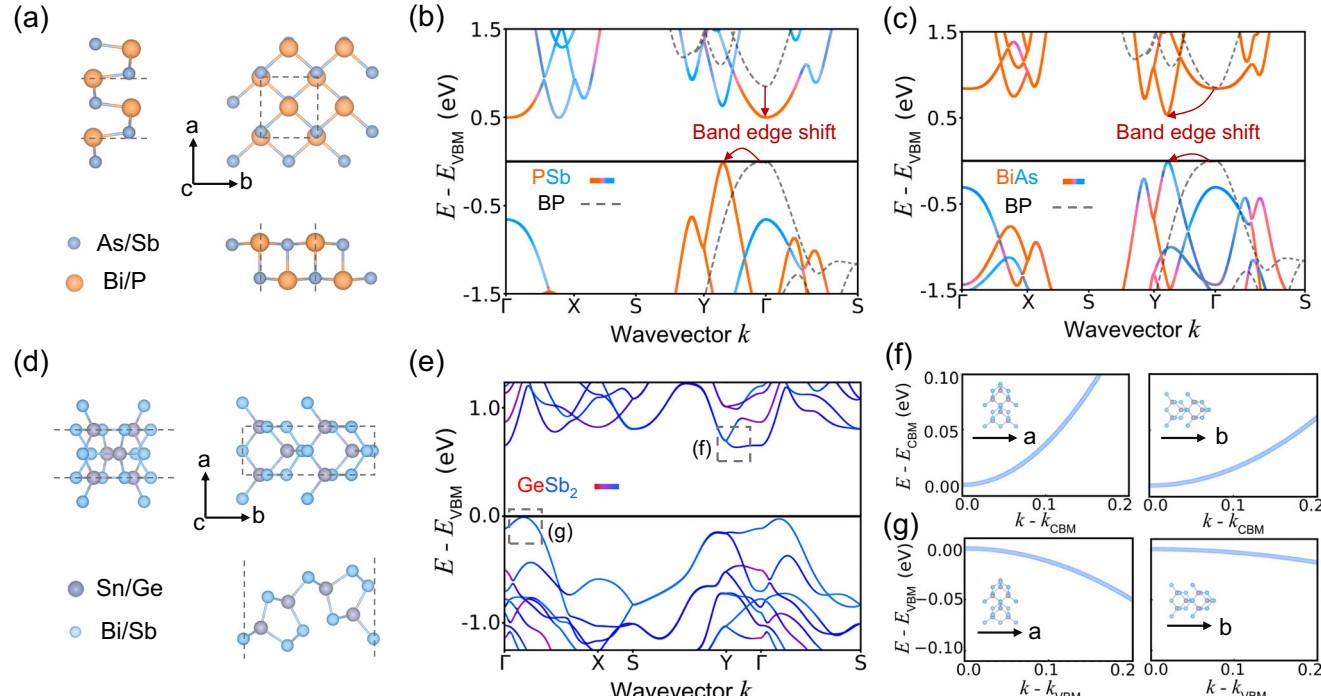

**Fig. 5 | Representative 2D materials with high carrier mobility. a** Crystal structures of group V binary compound AB (A = As/Sb, B = Bi/P), similar to black phosphorus. Electronic structure of (**b**) PSb and (**c**) BiAs, the gray dashed line is the electronic of black phosphorus (BP) and the red arrow describes the band edge shift after replacing phosphorus. The band color represents elemental contribution, indicated by colored text. **d** Crystal structures of group IV-V binary compound AB2 (A = Sn/Ge, B = Bi/Sb). **e** Electronic structure of GeSb2, and the gray boxes point out the band edges. **f, g** Detailed electronic structures of GeSb2 at band edges along different directions. More details about their band gap, effective mass, and deformation potential can be found in Table 1 and Supplementary Table 4. The structure files are provided in Supplementary Data 1.

approximation that was parameterized by Perdew, Burke, and Ernzerhof (PBE)[47]. In addition, based on the effective mass approximation, the mobility was computed by using the deformation potential theory[48,49] for 2D systems, which is expressed as Formula 1:

$$\mu_{2D} = \frac{e\hbar^3 C_{2D}}{k_B T m^* m_l^* \left(E_l^i\right)^2} \quad (1)$$

where $m^*$ is the average effective mass in two transport directions, $m_l^*$ and $E_l^i$ are the effective mass and deformation potential constant along the transport direction, and $C_{2D}$ is 2D elastic modulus, respectively. The automatic calculation workflow is demonstrated in Supplementary Fig. 8, and more computational details can be found in the Supplementary Information (SI).

Data collection and processing: The initial carrier mobility data for 178 2D materials were collected from published literature provided as the Supplementary Data. These data were then used as both training and testing sets. Meanwhile, we utilized two open-source 2D material databases, C2DB[50,51] and 2Dmatpedia[52], to serve as predicting sets. The source properties, i.e., bulk effective mass, are acquired from Materials Project[53]. Their element distribution can be seen in Supplementary Fig. 1, 2. Given the large range of mobility values and the difficulty in uniformly defining the carrier transport direction for different lattices, we employed two dimensionless quantities to describe the carrier mobility: the average carrier mobility $\bar{\mu}$ and mobility anisotropy $A$, as defined in Formula 2 and 3.

$$\bar{\mu} = log_{10}\left(\frac{\sqrt{\sum_i \mu_i^2}}{\mu_{Si}}\right) \quad (2)$$

$$A = log_{10}\left(\frac{\max\{\mu_i\}}{\min\{\mu_i\}}\right) \quad (3)$$

Where i can be two orthogonal transport directions and $\mu_{Si}$ represents the carrier mobility of silicon. These two quantities can describe both electron and hole mobility, which are noted as $\bar{\mu}_e$, $\bar{\mu}_h$, $A_e$ and $A_h$. The electron and hole mobilities in silicon[54] are 1331 cm² V⁻¹ s⁻¹ and 283.5 cm² V⁻¹ s⁻¹ respectively. To avoid the model placing too much emphasis on samples with large mobility or anisotropy, we logarithmically transformed all data.

**Reporting summary**

Further information on research design is available in the Nature Portfolio Reporting Summary linked to this article.

## Data availability

The carrier mobility data generated in this study are provided in the manuscript file, the Supplementary Information files, and Source Data files.

The data of 2D materials and bulk effective mass used in this study are available at public websites, C2DB[50,51] (https://cmr.fysik.dtu.dk/c2db/c2db.html), 2Dmatpedia[52] (http://www.2dmatpedia.org) and MP[53] (https://materialsproject.org/). The carrier mobility data for model training are provided in Supplementary Data 2. Source data are provided with this paper.

## Code availability

The codes to perform adversarial transfer learning and predict 2D carrier mobility are provided as Supplementary Code which are also available at https://github.com/XinYu-Chen98/Hybrid-ATL-and-expert-knowledge-for-materials-design[55].

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

## Acknowledgements

This work is supported by the National Key Research and Development Program of China (2022YFB3807200, 2022YFA5000703), Natural Science Foundation of China (22033002, T2321002, 22373013), Natural Science Foundation of Jiangsu Province, Major Project (BK20232012, BK20222007), Jiangsu Provincial Scientific Research Center of Applied Mathematics (BK20233002) and the Fundamental Research Funds for the Central Universities. The authors thank the computational resources from the Big Data Computing Center of SEU and the National Supercomputing Center in Tianjin.

## Author contributions

Q.Z. and J.W. conceived this work. X.C. proposed a hybrid transfer learning framework and wrote the code with guidance from S.L., Q.Z., and J.W., X.C. performed DFT calculations with guidance from Q.Z. and Q.C., X.C., Q.Z., and J.W. analyzed the data and co-wrote the manuscript, with input from the other authors.

## Competing interests

The authors declare no competing interests.
