## [Peer Review File · Nature Communications]

From bulk effective mass to 2D carrier mobility accurate prediction via adversarial transfer learningReviewers' Comments:

Reviewer #1 (Remarks to the Author):

The authors presented a transfer learning framework based on the idea of domain-adversarial training. This idea is used to tackle the issue of data scarcity in the 2D materials domain by transferring common knowledge learnable from bulk materials. 21 of the ~3,000 filtered candidates after screening with the machine learning model were validated using DFT calculations. The study would be more impressive if experimental validation is also presented.

From the machine learning perspective, the applied technique is not new, and the authors failed to cite Ganin et al. 2016 for introducing this idea. As 2D materials is a field that is out of my expertise, the only recommendation I can give regarding the publication of this paper is that this paper should be accepted only if the discovered candidates provide new insights to the 2D materials research community.

Some minor comments:

1. In figure 2, adding the case of modeling with expert features only would make the comparison more complete.
2. In the caption of figure 4, the SHAP plot is probably Fig. S6, not S7.
3. Please make sure the data and codes used in this paper are easily accessible to the readers for the sake of reproducibility and open science.

Reviewer #2 (Remarks to the Author):

The authors present a framework combining adversarial transfer learning and expert knowledge to predict carrier mobility of 2D materials using knowledge of bulk crystal structure. They use the method to screen out 21 new two-dimensional semiconductors with mobility higher than $10\text{Kcm}^2/\text{Vs}$ and reasonable bandgap, among 869 2D promising semiconductors selected from open source materials databases.

I have a major comment and criticism: The authors do not make any comparison in terms of effectiveness and cost of using their proposed method with alternatives based on other machine learning techniques or other rule-based algorithms. Just as an example, would it be possible to select a smaller number of initial 2D materials (from the 869) from the database based on some bandstructure property or on some simple calculation using as set of bandstructure properties as input, that could enable to predict a higher mobility?

As an additional verification, can they compare the computational cost of their proposed machine learning method (including the cost of training) with the cost of running a DFT calculation on all the initial 869 2D materials? or with the cost of running a simplified (less precise, but faster) DFT calculation on all the initial 869 2D materials? We all know that a small improvement in precision of a DFT calculation comes at a significant computational cost.

In conclusion, I think that in order for this paper to be considered for publication the authors should justify quantitatively why this ML-based screening method is preferable with respect to algorithmic and explainable alternatives, that have the intrinsic advantage to be more easily verifiable by different groups.

Response to Reviewer #1

The authors presented a transfer learning framework based on the idea of domain-adversarial training. This idea is used to tackle the issue of data scarcity in the 2D materials domain by transferring common knowledge learnable from bulk materials. 21 of the ~3,000 filtered candidates after screening with the machine learning model were validated using DFT calculations. The study would be more impressive if experimental validation is also presented.

Reply:

We appreciate the positive feedback and helpful comments provided by the reviewer. This work focuses on machine learning and theoretical computation, so conducting additional experimental validation is beyond the scope of our research. Moreover, as a pure theoretical research group, such experimental validation is also beyond our capability. However, we have still investigated experimental progress related to the materials we have identified. It is exciting to note that among 21 screened materials, In_4Se_3 and Nb_2SiTe_4 have already been successfully synthesized and experimentally verified to exhibit excellent charge carrier transport capabilities at room temperature (ACS Nano 2019, 13, 10705-10710; ACS Nano 2022, 16, 8107-8115). This suggests that the candidates we have identified do have large potential to be synthesized in experiment and hold high charge carrier mobility.

Revision made:

We added a discussion on the progress of experimental synthesis of these materials in page 12 in the revised manuscript:

'Note that among the screened 21 materials, some have already been synthesized or even been experimentally validated to have high mobility, such as In_4Se_3 and Nb_2SiTe_4 .^{45-47'}

From the machine learning perspective, the applied technique is not new, and the authors failed to cite Ganin et al. 2016 for introducing this idea. As 2D materials is a field that is out of my expertise, the only recommendation I can give regarding the publication of this paper is that this paper should be accepted only if the discovered candidates provide new insights to the 2D materials research community.

Reply:

1. Novelty of machine learning methods: We apologize for failing to cite Ganin's paper and acknowledge that our work shares some similar ideas with theirs. However, we do not agree that this diminishes the novelty of our research. In Ganin's study, they used domain adversarial neural networks to capture common features and achieve knowledge transfer across different images. However, in the context of material design, conducting transfer learning is more challenging due to the complexity of material structures and properties. The novelty of our material-oriented framework lies in achieving simultaneous cross-materials and cross-property knowledge transfer, which greatly expands the application scope of transfer learning in materials design as shown in Fig. R1. Below we will give more comprehensive discussion.

The challenge of applying this idea to materials, in comparison to images, stems from the large domain difference in different materials and the lack of a comprehensive representation for all materials, like pixels for all images. The effectiveness of transfer learning largely relies on the

similarity between the target and source domains. It is well known that when the domain differences are significant, transfer learning may not provide helpful insights and can even result in poorer predictions, a phenomenon commonly referred to as negative transfer (Fig. R1). In fact, the domain differences in materials are usually large, arising from two primary factors. First, materials can possess highly diverse structures and compositions that make capturing their similarities challenging. Second, materials properties are complex, and describing different properties may require specific and distinct features. In such scenarios, when attempting to transfer knowledge from big datasets to small datasets with different materials and different properties, negative transfer is highly likely to occur. To mitigate this issue, the introduction of domain adversarial training in the realm of materials is highly valuable, which helps us capture the similarity of different materials.

Figure R1. Dependence of domain difference and transfer learning effectiveness. In our framework, we incorporate adversarial training and expert knowledge to capture similarities between distinct materials and describe the uniqueness of the target properties, respectively, which achieves cross-materials and cross-property knowledge transfer, and thereby expands the applicability of transfer learning.

However, relying solely on domain adversarial training is still insufficient for material applications. Unlike images, where pixels contain most of the necessary information, materials possess diverse and unique structures and properties that cannot be fully captured by a single representation. In other words, even though adversarial training can extract common features for different materials, the extracted features are also ineffective if the original features cannot effectively describe the target materials and properties. This limitation is particularly pronounced for 2D materials, as many of their novel properties stem from their distinctive structures. Consequently, it becomes crucial to consider additional information that can effectively describe the unique features of materials in the target domain. Therefore, we have extended beyond the original concept to develop a hybrid framework that incorporates features from expert knowledge, for example, we add the lattice symmetries that can significantly impact phonon vibration modes and phonon-electron scattering, which are critical factors influencing carrier mobility. This enhancement compensates for the lack of unique 2D characteristics based only on adversarial training, while the inclusion of human knowledge on carrier mobility facilitates more robust and interpretable predictions. Furthermore, the additional features can be substituted for different applications. This additional aspect significantly enhances the reliability and flexibility of our approach in the context of material design.

To summarize, our work emphasizes the handling of material complexity, and represents a

pioneering approach that combines adversarial training with expert knowledge, thereby expanding the application scope of transfer learning in materials design. This establishes transfer learning as a widely applicable strategy for addressing data scarcity in AI-assisted material design. Therefore, we firmly believe that our framework is of great novelty and holds promise for advancing the field.

2. New insights to the 2D materials: The carrier mobility of 2D materials is a critical parameter for semiconductor device applications, and materials with high carrier mobilities are promising candidates for the continued scaling of transistors. However, estimating carrier mobility is a complex and time-consuming process, often requiring extensive density functional theory (DFT) calculations. Currently, there is no efficient and accurate method for large-scale prediction of carrier mobility in 2D materials, which poses a significant challenge to materials design and discovery. Our work presents the first practical tool to realize such rapid 2D carrier mobility prediction with DFT accuracy.

To enable the rapid and accuracy prediction of 2D carrier mobility, we have innovatively incorporated expert knowledge and interpretable models into our approach. This has allowed us to gain a comprehensive understanding of the critical factors that influence 2D carrier mobility and generate new and exciting results. For instance, we have identified the covalency of element bonding as a key factor affecting carrier mobility in 2D materials. Our study has also revealed, for the first time, that in-plane mirror symmetry should be considered as a potential influencing factor to carrier mobility. Building upon these findings, we have proposed a method to improve the carrier mobility of 2D materials through element substitution and symmetry manipulation. By substituting elements with stronger covalent bonding and manipulating their symmetry, we can enhance the valence electron distribution near the Fermi level, leading to higher carrier mobility.

In summary, the significant contribution of our work to 2D materials community is that it provides a practical tool for the fast prediction of their carrier mobility and gives guidance to design high mobility 2D materials.

Revision made:

We added Ganin et al. 2016 (Domain-Adversarial Training of Neural Networks, J. Mach. Learn. Res. 2016, 17, 1-35) as ref.29 to give their credit for the idea of domain-adversarial training they firstly introduced in the realm of computer vision:

'The domain adversarial training method was first introduced in the realm of computer vision to learn common knowledge between different images.³⁰ Here, we employ similar adversarial training concept to acquire common knowledge between different materials, meanwhile, we incorporate a priori knowledge of chemistry to better describe the uniqueness of material property.'

(Page 3)

We added Fig. R1 as TOC diagram to clarify the challenges the effort we made to extend the applicability of transfer learning in materials design. Additional discussions are also added to clarify the novelty of this work:

'Transfer learning is a machine-learning algorithm that can improve the performance of learner on small dataset (target domain) by transferring the knowledge from different but large dataset (source domain). It has been considered as a very promising approach to address the data scarcity challenge in ML-assisted material design.^{14,15} For example, Liu et al. successfully predicted phonon properties of bulk semiconductors by training on 1245 electronic bandgaps and finetuning on 124

phonon bandgaps.¹⁶ Similarly, Li et al. accurately predict the formation energy of perovskite oxides by training on 5329 spinel oxides and finetuning on 855 perovskite oxides.¹⁷ However, current transfer learning applications are either between different properties with the same materials (cross-property), or between different materials with the same property (cross-material).¹⁸⁻²⁴ This is owing to that the effectiveness of transfer learning is closely related to the difference between source and target domain, and if the domain difference is too large, it will not be effective and may give poorer predictions, i.e., negative transfer.²⁵ (Page 2)

‘In practical applications, the problem of data scarcity becomes even more pronounced, as our extensive databases typically only cover fundamental properties of widely-used materials. Yet, our focus is often on a particular category of materials, for which we strive to predict their more complex properties. Carrier mobility in atomically thin 2D semiconductors is such a typical example. 2D materials with suitable bandgap and high carrier mobility are expected to facilitate the continued transistor scaling.^{26,27} However, the evaluation of carrier mobility is a costly process that often requires extensive density functional theory calculations, as a result, the available data is very limited.^{28,29} Additionally, 2D materials themselves are recent additions to the material family which also lacks sufficient data. In contrast, bulk materials have been studied for a much longer period and have rich data available, including diverse properties, in which the effective mass is believed to closely related to carrier mobility. Naturally, we hope to utilize bulk effective mass data to enhance the prediction of 2D carrier mobility. However, owing to the diversity of 2D material structures and the complexity of their properties, simultaneous cross-material and cross-property transfer learning poses a greater challenge.’ (Page 3)

‘However, with the help of adversarial training, common knowledge between bulk and 2D materials are captured, and negative transfer is alleviated. It is also essential to recognize that many appealing properties of 2D materials stem from their unique structure. Therefore, additional features based on expert knowledge describe their special structures, which complements the knowledge acquired from the bulk materials and leads to more accurate predictions.’ (Page 8)

Some minor comments:

1. In figure 2, adding the case of modeling with expert features only would make the comparison more complete.

Reply:

Thank the reviewer for this advice, the model with expert features only has been added in Fig. 2(a) (Fig. R2) for comparison, which shows slightly improvement compared to baseline and the hybrid approach remains to be the best model.

Figure R2. Performance comparison of different learning method. The baseline (dash line) is set to direct prediction from MAGPIE without transfer learning.

2. In the caption of figure 4, the SHAP plot is probably Fig. S6, not S7.

Reply: The caption has been corrected.

3. Please make sure the data and codes used in this paper are easily accessible to the readers for the sake of reproducibility and open science.

Reply: The code and training data are provided as review-only files for now, with full open-access availability at Github upon acceptance of the paper.

Response to Reviewer #2

The authors present a framework combining adversarial transfer learning and expert knowledge to predict carrier mobility of 2D materials using knowledge of bulk crystal structure. They use the method to screen out 21 new two-dimensional semiconductors with mobility higher than $10\text{Kcm}^2/\text{Vs}$ and reasonable bandgap, among 869 2D promising semiconductors selected from open source materials databases.

Reply: We thank the reviewer for the critical comments to further improve our manuscript. Below is our reply to the comments.

I have a major comment and criticism: The authors do not make any comparison in terms of effectiveness and cost of using their proposed method with alternatives based on other machine learning techniques or other rule-based algorithms. Just as an example, would it be possible to select a smaller number of initial 2D materials (from the 869) from the database based on some bandstructure property or on some simple calculation using as set of bandstructure properties as input, that could enable to predict a higher mobility?

Reply:

Thank the reviewer very much for his/her constructive suggestions. We did discuss the effectiveness of our model predictions in the original manuscript and supporting materials. For instance, we compared the accuracy of our method with other transfer learning methods in Fig. 2, and in Fig. S5, we compared it with some classical machine learning algorithms. Additionally, in Fig. 3, we compared the accuracy of our method and density functional theory (DFT) based mobility calculations. These results demonstrate that our approach provides comparable DFT accuracy, and is more accurate than other machine learning alternatives.

In terms of computational cost, it is indeed valuable to compare and highlight the speed advantage of our machine learning approach. As depicted in Fig. R3, our method is capable of providing hundreds of predictions within seconds, which is significantly faster by 5 orders of magnitude compared to DFT-based mobility calculations. Furthermore, when compared to other transfer learning models, the introduced adversarial network is only utilized during model training and does not affect the prediction speed. The model training process typically takes a few hours, which is comparable to calculating the mobility of a single material based on deformation potential approximation. Since model training only needs to be performed once, this speed can still be considered very fast and will not slow down the mobility prediction in practice.

To screen a smaller size of material data based on the band structure or add the band structure as an input feature, as the reviewer suggested, we first need to calculate the band structure. However, calculating band structure is also time-consuming and significantly slower than TL mobility prediction, as shown in Fig. R3. The primary goal of utilizing machine learning for material design is to offer a rapid prediction tool that facilitates swift screening of a vast materials space. In this regard, a favorable machine learning framework should be DFT-free, just like our presented approach. Incorporating complex features into the prediction process would significantly hinder its efficiency. Although we did observe that adding the band structure feature, such as band gap, as an input feature in our early tests (Fig. R4) can improve the prediction accuracy, the enhancement is not substantial. It is not a favorable trade-off to invest thousands of times more computational resources for such marginal accuracy improvement. Therefore, we do not pursue further band structure-based material screening or incorporate band structure features in our current approach.

Figure R3. Time cost comparison of prediction with trained transfer learning model (TL), DFT-based band structure calculation (Band) and deformation potential calculation (Full DP), the error bar indicates the standard deviation on calculating different materials and the dashed line is the time cost to train TL model. TL predictions are performed on single core of Intel Xeon Gold 5218, TL model training is performed on Nvidia RTX 2080Ti and single core of Intel Xeon Gold 5218, and the DFT calculations are performed on 48 cores of Intel Xeon Gold 6248R.

Figure R4. Prediction accuracy of electronic mobility with and without band gap as input feature.

Revisions made:

We added Fig. R3 as Fig. S10 in the revised supplemental information, and a discussion on the computational cost of different method in the revised manuscript:

“Compare to DFT-based mobility calculation, our approach is five orders of magnitude faster as shown in Fig. S10.” (Page 10)

‘Figure S10 provides a comparison of the time costs associated with TL-based mobility prediction, DFT-based band structure calculation and deformation potential calculation. Our TL

model demonstrated computational speed improvements of four to five orders of magnitude when compared to DFT calculations. Additionally, the overall training time for our model was found to be similar to the time required for calculating the deformation potential of a single material, and we don't need to retrain the model when perform predictions. It is worth noting that the time cost of DFT is strongly dependent on the cell size and atom number, resulting in considerable variation in computation times across different materials, which can be seen from the large error bar. In contrast, the network complexity of TL is fixed, which ensures that its prediction speed will not be significantly slowed down, regardless of the structural complexity of the material being predicted.' (Page S12)

As an additional verification, can they compare the computational cost of their proposed machine learning method (including the cost of training) with the cost of running a DFT calculation on all the initial 869 2D materials? or with the cost of running a simplified (less precise, but faster) DFT calculation on all the initial 869 2D materials? We all know that a small improvement in precision of a DFT calculation comes at a significant computational cost.

Reply: Running DFT-based carrier mobility calculations on all 869 2D materials is extremely expensive. Even calculating the selected 21 2D materials requires thousands of CPU core hours. Therefore, conducting such calculations for all 869 materials is impractical. However, we can use the calculated results from the 21 materials to estimate the time required for calculating all 869 materials. As shown in Fig. R5, the overall time cost for obtaining the mobility of 869 2D materials, including model training, is three orders of magnitude faster compared to DFT-based mobility calculations. This significant improvement in speed cannot be achieved by solely running simplified DFT calculations. Importantly, the majority of the time is spent on model training, which only needs to be done once. This indicates that as the scale of 2D materials expands further, the efficiency advantage of TL mobility prediction over DFT-based mobility calculations will continue to grow.

Figure R5. Estimated time cost of different calculating method for all 869 stable 2D materials. The insert shows the proportions of time cost of training and prediction phases in transfer learning.

In conclusion, I think that in order for this paper to be considered for publication the authors should justify quantitatively why this ML-based screening method is preferable with respect to algorithmic and explainable alternatives, that have the intrinsic advantage to be more easily verifiable by different groups.

Reply: Thank the reviewer very much for his/her constructive comments. As discussed above, compared to DFT-based mobility calculation, our ML-based model is significantly faster, which enables rapid prediction and massive screening of a large number of materials. Compared to other machine learning-based alternatives, our approach offers the advantage of providing more accurate predictions even with limited available data. This is achieved through the combination of adversarial transfer learning and expert knowledge, effectively leveraging the knowledge learned from bulk effective mass. Compared to other transfer learning methods, our approach is more effective in capturing the similarities between different materials and providing a more complete description of the target property. As a result, it significantly improves prediction accuracy and enables simultaneous cross-material and cross-property knowledge transfer. In summary, our method has several intrinsic advantages over DFT-based and ML-based alternatives, including better running efficiency, prediction accuracy, data requirement, model interpretability and usability.

1. Running efficiency and prediction accuracy: Compared to DFT-based mobility calculations, our approach is significantly faster by 4-5 orders of magnitude with DFT accuracy. Compared to other machine learning approaches, our model gives significantly more accurate prediction as summarized in Table R1. Among all machine learning based alternatives, our model gives the highest prediction accuracy reaching 90%, despite the training cost is slightly higher, it is still significantly faster than conducting DFT calculations.

Table R1. Comparison of running time and prediction accuracy, the running time refers to the estimated time for predicting 1000 2D materials, the prediction accuracy is evaluated by R^2 score compared to DFT-based calculations.

Method	Accuracy	Time (seconds)
DFT-deformation potential	1.00	$\sim 10^7$
Hybrid transfer learning (this work)	0.89	$\sim 2 \times 10^3$ (train) / ~ 10 (predict)
Feature-based transfer learning	0.37	$\sim 5 \times 10^2$ (train) / ~ 10 (predict)
Scratch multilayer perceptron	0.55	$\sim 2 \times 10^2$ (train) / ~ 10 (predict)
LASSO	0.61	~ 10 (train) / $\sim 10^{-1}$ (predict)
Kernel Rigid Regress	0.60	~ 10 (train) / $\sim 10^{-1}$ (predict)

2. Data requirements and model interpretability: In comparison to other ML-based predictions, our approach can provide more accurate predictions with fewer training samples as summarized in Table R2. To achieve this, we use adversarial training to learn knowledge from bulk effective mass and an engineered unique feature set to characterize unique 2D structures. Moreover, we have employed interpretable machine learning to analysis the contribution of each input features,

which provides deeper insights into the decisive factor of 2D carrier mobility.

Table R2. Comparison of prediction accuracy for different machine learning method with different sample size.

Number of materials	Method	Accuracy
222	Hybrid transfer learning (this work)	0.89
	Scratch multilayer perceptron	0.55
	LASSO	0.61
	Kernel Rigid Regress	0.60
100	Hybrid transfer learning (this work)	0.72
	Scratch multilayer perceptron	0.47
	LASSO	0.51
	Kernel Rigid Regress	0.56
50	Hybrid transfer learning (this work)	0.52
	Scratch multilayer perceptron	0.30
	LASSO	0.38
	Kernel Rigid Regress	0.37

3. Usability across different materials and properties: Transfer Learning is a machine learning algorithm that enhances the prediction on a small dataset (target domain) by transferring knowledge from a different but large dataset (source domain). This approach has shown great potential in addressing the challenge of limited material data in AI-assisted material design. However, its effectiveness heavily depends on the similarity between the source and target domains. Significant domain differences can lead to negative transfer, resulting in poorer predictions and restricting the application scope of TL as shown in Fig. R1. Conventional transfer learning methods are typically effective in transferring knowledge across datasets with minor domain differences, such as different properties for the same materials or identical properties for different materials. Our model expands the scope of transfer learning by enabling simultaneous cross-material and cross-property knowledge transfer. This is achieved through the utilization of adversarial training to capture similarities between distinct materials and the incorporation of specific expert knowledge to describe the target property.

In summary, our framework enables the rapid prediction of 2D carrier mobility with DFT accuracy, which was a highly challenging task in the field of 2D materials. Furthermore, this framework serves as a universal tool that extends the applicability of transfer learning to diverse materials and properties, effectively addressing the issue of limited data availability. By leveraging knowledge from various materials and allowing flexible expert feature selection, our framework facilitates accurate predictions even for materials with complex properties and minimal available

data. This provides a practical and versatile solution for handling data scarcity in materials research.

Revisions made:

We added Table. R1 as Table. S3 in the revised supplemental information, and a discussion on the overall accuracy and efficiency compared to other DFT-based and ML-based alternatives:

‘The success of this method lies in that the adversarial training can effectively captures common features between different materials while incorporating expert knowledge can better describe the uniqueness of target property. This is the first framework for realizing simultaneous cross-material and cross-property transfer learning, which greatly broadens the domain difference ranges of transfer learning and establishes it as a widely applicable strategy for addressing data scarcity in ML-assisted material design.’ (Page18)

Response to Reviewer #1

I apologize for the unclear notes in my previous review, but I do not intend to change my opinion about the paper even after the revision. I believe that there is a misunderstanding of my comment and I find it necessary for me to clarify my opinion in more detail.

First of all, I did not intend to discredit the contribution that the authors introduced adversarial transfer learning plus expert knowledge to the 2D materials domain. My comment is that from the machine learning perspective, there has been a rich set of studies that address the idea of using adversarial training to improve transfer learning performance (e.g., avoid negative transfer). My intention of mentioned Ganin's paper is just to point out this reality to the authors, especially because the probably setup in this paper is closely related to domain adaptation, which is what Ganin et al. targeted in general. As a result, I believe that in order to evaluate the level of contribution of this paper to the materials science community, it is important to focus their discussion on the physical insights gained from their specific design of adversarial transfer learning approach applied to 2D materials. Unfortunately, I am not an expert in this field of 2D materials, so I cannot verify the value of their discussion accurately. In short, I am trying to point out that if the authors are trying to claim great novelty of their paper purely based on their introduction of adversarial transfer learning and expert knowledge to this application by just showing an improvement of prediction accuracy on test data, this is not sufficient for a journal like Nature Communications. Some examples of on-going progress of adversarial transfer learning in the machine learning field:

1. Review paper

https://link.springer.com/chapter/10.1007/978-981-19-7584-4_10

2. Example of theoretical progress

<https://proceedings.neurips.cc/paper/2021/file/d3aeec875c479e55d1cdeea161842ec6-Paper.pdf>

3. Example of application in materials informatics

<https://pubs.acs.org/doi/abs/10.1021/acs.iecr.0c02398>

Reply:

We appreciate the constructive comments from the reviewer, which have aided us in clarifying the contribution of our work. We would like to emphasize that this study is not purely oriented to improve transfer learning performance, but represents an interdisciplinary research effort that bridges machine learning and material science. Overall, this work can be regarded as applied AI research, which is essential alongside theoretical AI research, and one popular topic is AI for materials. Several related studies have been published in top journals. For instance, Lu et al. employed ensemble learning to rapidly discover stable lead-free hybrid organic-inorganic perovskites (Nat. Commun. 9, 3405, 2018). Batra et al. utilized supervised learning to accurately predict water stability in metal-organic frameworks (Nat. Mach. Intell. 2, 704-710, 2020). Moon et al. employed active learning to guide the discovery of a champion four-metal perovskite oxide for electrocatalysis (Nat. Mater. 23, 108-115, 2024). These efforts focus on leveraging AI to address real-world challenges in materials design, and there are many such endeavors. Similarly, the importance of this work lies in optimizing transfer learning methods for complex material systems, allowing for accurate prediction of intricate material properties, such as 2D carrier mobility.

From the machine learning perspective, as discussed in previous response, we acknowledge that the conception of adversarial transfer learning is inspired by Ganin's pioneering work, and the

primary contribution of this work lies in modifying such an idea and incorporating additional expert knowledge to materials research. This enables the knowledge transfer across different materials and properties simultaneously, improving the accuracy and the interpretability of the predictions for materials with complex structures and properties. Such method provides an effective solution to predict intricate material properties with limited data.

From the perspective of 2D materials, carrier mobility stands as a crucial property within the semiconductor industry, as it directly influences the speed of charge transfer and device responsiveness. Therefore, the rapid and precise prediction of 2D carrier mobility holds significant value in the quest for high-performance 2D semiconductors. Due to the intricate electron-phonon interaction involved in charge transport, current computational methods for estimating carrier mobility are exceedingly complex and time-consuming. Furthermore, owing to the scarcity of carrier mobility data, a machine learning-based tool for rapid 2D carrier mobility prediction is still lacking. Leveraging the proposed hybrid transfer learning approach, this work achieves accuracy comparable to density functional theory (DFT)-based mobility calculations, but with a speed five orders of magnitude faster, thus offering a valuable tool for 2D materials design.

We have revised the manuscript to enhance clarity regarding the contributions from both the machine learning and materials science domains. Additionally, we acknowledge and appreciate the ongoing advancements in adversarial transfer learning within the machine learning field, as they contribute to refining our methodology. These advancements have been incorporated into the discussion section of our work to provide guidance for future improvements.

Revision made:

We have modified the abstract and the main text to clarify the contribution of this work in terms of machine learning methodology as follows:

‘This work achieves transfer learning in simultaneous cross-property and cross-material scenarios, providing an effective tool to predict intricate materials properties with limited data.’(Abstract)

‘Notably, such mobility prediction only necessitates crystal structures as input, yet maintains accuracy comparable to DFT calculations but at a speed five orders of magnitude faster. Moreover, 21 2D semiconductors with ultra-high carrier mobility far exceed silicon have been screened out from 4266 candidates.’ (Page 17, contribution regarding to 2D materials)

‘The success of this method lies in incorporation of adversarial training and expert knowledge, which effectively captures similarity among diverse materials while also characterizing the distinctive attributes of target materials and properties. Therefore, it facilitates simultaneous cross-material and cross-property transfer learning, enhancing the predictive capabilities and reliability of the model. This study provides a widely applicable strategy for addressing data scarcity in ML-assisted material design.’ (Page 17, contribution regarding to material-oriented machine learning method)

We have also incorporated on-going advancements in adversarial transfer learning within the machine learning domain as ref.41-42.

‘It may require an improved adversarial training approaches or borrowing generative adversarial network methods from inverse design.’⁴¹⁻⁴³ (Page 17)

In addition, I would like to note that the idea of including expert knowledge through concatenation of descriptors is very common. The more exciting aspect of including expert

knowledge in such machine learning models would be if the authors can show some evidence of new insights through such combination of machine learned descriptors and conventional expert descriptors, say using some explainable AI techniques. By the way, it is great that the authors added the expert-knowledge-only model as a comparison.

Reply:

We agree it is a common way in machine learning research to concatenate descriptors. The crucial aspect lies in understanding why adversarial transfer learning model requires additional expert knowledge and what specific expert knowledge should be incorporated. We have included a discussion on this aspect to provide clarity.

The necessity for expert knowledge stems from the fact that adversarial training only captures common features across different materials, lacking the uniqueness of target materials and properties. Therefore, incorporating specific expertise becomes vital. In the case of 2D carrier mobility, we have chosen to include lattice symmetry, crystal geometry, and electronic properties to describe the unique 2D structures and their electronic behavior.

As recommended by the reviewer, we have already incorporated explainable AI techniques, such as SHAP, to extract insights from such combination of machine-learned descriptors and expert descriptors, as illustrated in Fig. 3 and 4. These analyses evaluate the contributions of different feature sets across various tasks and also point out critical factors that determine 2D carrier mobility. This showcases how machine-learned descriptors and expert descriptors complement each other, leading to more robust and rational predictions. It also emphasizes the importance of adding the proper expert knowledge regarding to the target properties and materials.

Finally, we thank the reviewer for the previous suggestion to add expert-knowledge-only model as a comparison.

Revision made:

We added a discussion on the necessity for incorporating expert knowledge:

'The necessity of incorporating expert knowledge lies in the fact that the adversarial approach only ensures the extraction of common knowledge, but it lacks the description of the uniqueness of target materials and their properties. This is particularly critical in cases like 2D materials, where many interesting properties stem from their unique structures. Therefore, in addition to the features extracted by transfer learning, we add features from lattice symmetry, crystal geometry, and electronic properties to describe the unique 2D structures and their electronic behavior.' (Page 5)

Finally, I would like to clarify that my comment of adding experimental validation is just an example of showing more evidence of their contribution than just claiming an improvement on prediction accuracy. It is not meant to be necessary at all.

Reply:

Thank you for your feedback and previous suggestions regarding the need for experimental validation. We also aimed to convey the broader impact of our work and agree that emphasizing the potential applications of the materials identified through our study enhances the significance of our findings. Therefore, we have opted to maintain the discussion on experimental validation.

Response to Reviewer #2

The authors have properly addressed all the issues and the comments that I posed in my first review. The material that they have added to the main manuscript and to the supplementary

information is useful to clarify the main issue I raised (the need to have a comparison in terms of effectiveness and cost between using their proposed method and alternatives based on other machine learning techniques or other rule-based algorithms).

The authors have included comparisons of computational costs of the different methods, as I was asking for. Such comparisons are convincing and can be useful to the reader.

For these reasons, I consider the manuscript in its present form to be suitable for publication on Nature Communication.

Reply:

We sincerely appreciate your constructive feedback on our manuscript and the thorough review you provided. Your comments guided us in addressing the key issue regarding the need for comparisons between our proposed method and alternative approaches. We are grateful for your support and endorsement of our work for publication.